# Etiology of IBD—Is It Still a Mystery?

**DOI:** 10.3390/ijms232012445

**Published:** 2022-10-18

**Authors:** Anna Kofla-Dłubacz, Tomasz Pytrus, Katarzyna Akutko, Patrycja Sputa-Grzegrzółka, Aleksandra Piotrowska, Piotr Dzięgiel

**Affiliations:** 12nd Department of Paediatrics, Gastroenterology and Nutrition, Faculty of Medicine, Wroclaw Medical University, 50-367 Wroclaw, Poland; 2Division of Anatomy, Department of Human Morphology and Embryology, Wroclaw Medical University, 50-367 Wroclaw, Poland; 3Division of Histology and Embryology, Department of Human Morphology and Embryology, Wroclaw Medical University, 50-367 Wroclaw, Poland

**Keywords:** IBD, innate immune response, adaptive immune response, 12/23 cytokines, IL10, MMPs

## Abstract

Inflammatory bowel diseases (IBD), including colitis ulcerosa and Crohn’s disease, are chronic diseases of the gastrointestinal tract for which the cause has not been fully understood. However, it is known that the etiology is multifactorial. The multidirectional network of interactions of environmental, microbiological and genetic factors in predisposed persons lead to an excessive and insufficiently inhibited reaction of the immune system, leading to the development of chronic inflammation of the gastrointestinal walls, the consequence of which is the loss of the function that the intestine performs, inter alia, through the process of fibrosis. Detailed knowledge of the pathways leading to chronic inflammation makes it possible to pharmacologically modulate disorders and effectively treatthese diseases. In this review, we described the primary and adaptive immune system response in the gut and the known immune pathogenetic pathways leading to the development of IBD. We also described the process leading to intestinal tissue fibrosis, which is an irreversible consequence of untreated IBD.

## 1. Introduction

Inflammatory bowel disease (IBD) is a group of chronic inflammatory diseases of the gastrointestinal tract, which include Crohn’s disease (CD), ulcerative colitis (UC), and the less common IBD-unclassified (IBD-U), including forms with an ambiguous clinical picture.

The incidence of IBD is increasing in the general population, which makes it a significant clinical challenge. It is estimated that the global prevalence of IBD is currently close to 90 cases/100,000 people. However, attention should be paid to the graphical differentiation and trends in changes in the incidence and frequency of IBD in different regions of the world [1]. In Europe the prevalence of IBD is the highest, at 505/1,000,000 for UC and 322/100,000 for CD, respectively. A slightly smaller population lives in North America, where it reaches 248/100,000 for UC and 318/100,000 CD. In developing countries, the number of new cases has been growing dynamically [2,3,4].

The costs of this increase are becoming noticeable on many levels, including diagnostic and therapeutic as well as social and economic, which stimulates research centers to look for new forms of effective therapies. The issue remains difficult because the etiology of IBD is multifactorial and the mechanisms of persistent immune response are regulated at multiple levels, being the result of the interaction between genetic, environmental and immune factors [5].

The purpose of this review is to present the known immune factors that influence the development of IBD and the relationships between them. Insight into innate and adaptive immunity impaired in IBD provides a comprehensive picture of the multifactorial etiology of the disease. The most explored elements of the puzzle of interactions are the breakage in the protective barrier of the intestinal epithelium (primary defense) and the activation of pro-inflammatory elements of the specific response, mainly the cytokines 12/23, with simultaneous suppression of the anti-inflammatory effect of the intercellular transmitters.

## 2. Primary Defence Line in IBD

The digestive tract, due to its structure and function, is a place of constant interaction between the external components and the immune system so the integrity of intestinal border stays crucial in homeostasis between them. The main elements of the border are the epithelial cells connected by effective cell junctions, the mucosal layer covering them and the individual pattern of the microbiome. The microbiome with its diversity has an impact on the immune response but the relation is two-sided. The immune cells are able to modify the composition of the bacterial flora of the digestive tract as well. The stability of epithelium determines the degree of penetration of environmental antigens through the intestinal wall and farther their presentation to the cells of the immune system.

The composition of the gut flora seems to play a key role in the development of IBD. It is also known that disruption of microbiota (dysbiosis) is a cause of the exacerbation of the disease and determines the severity of the inflammation. The predominance of pathogenic strains, coming from overexposure to antibiotics and industrial chemicals, contributes to the increase in the incidence of IBD.

Although the pattern of the microbiome is highly individualized and the possibilities of modifying it are still small, some kinds of pathogens are more frequently linked with a higher risk of initiation and progression of IBD. One of these is a virulent serotype of *Clostridium difficile*, a gram-positive, toxin-producing bacteria. Although the strict connection between overgrowth of this microbe and pathogenesis of IBD has not been established, the association between infection and IBD has been observed. Even 10% of patients may reveal positive tests for *C. difficile* at the time of diagnosis and up to 19% of the IBD patients with an exacerbation of the disease [6]. *C. difficile* by producing its toxins, (Toxin A and B) that bind to epithelial cells, activates production of the proinflammatory cytokines such as the tumour necrosis factor (TNF) and interleukins-6, -8, -1β, that disturb the integrity of the intestinal border [7]. Infectious agents other than *C. difficile* are also taken into consideration in pathogenesis of IBD. Although it is not easy to settle whether inflamed mucosa promotes the infection or pathogens initiate the inflammatory answer first, the imbalance in the integrity of epithelium barrier in the relation to the infection seems to be indisputable. *Campylobacter jejuni*, *Salmonella typhi*, enteroinvasive and enteroadherent serotypes of *Escherichia coli* may be engaged in initiation of IBD. Additionally, bacteria from the *Mycobacterium* group: (*M. avium* and *M. paratuberculosis*) were suspected in the pathogenesis of IBD [8,9]. While the mechanism by which *Salmonella* spp. and viruses such, as rotavirus and adenovirus, contribute in the development of IBD is unknown, *Campylobacter* spp. seems to increase translocation of microbiota through intestinal wall as a result of increased lipid raft-mediated transcytosis and tight junction depletion. The bacteria from the *Mycobacterium* group and invasive or adherent *Escherichia coli* lead to chronic infection for a change and persistent stimulation of the immune system [10].

Although a protective pattern of the microbiome has not been described and a specific type of IBD-promoting dysbiosis has not been discovered, there is indirect data supporting this association.

The effective use of antibiotics in the treatment of IBD in acute patients confirms the influence of the microbiome on the course of the disease. The randomized, controlled PRASCO trial [11] compared two groups of IBD patients treated one with a cocktail of antibiotics like (amoxicillin, vancomycin, metronidazole, doxycycline/ciprofloxacin) and intravenous corticosteroids (IVCS) and the other with single use of IVCS. Disease activity measured by paediatric ulcerative colitis activity index—PUCAI was lower in the antibiotic group than in the IVCS group. The difference was statistically significant. However, in the study, the therapeutic answer was not linked to the specific species (analyzed using the 16s rRNA gene and metagenome).

Moreover, clinical studies on the effects of therapeutic diets on the composition of the bacterial flora and further on the severity of inflammation, indicate that these diets could potentially reduce dysbiosis and limit the inflammation. The exclusive enteral nutrition (EEN), which is currently the standard of care in the induction of remission in patients with CD, influences the composition of the gut microbiome. In clinical trials comparing the efficacy of EEN treatment versus steroids, a higher clinical and histological response, as well as a modulating effect of this treatment on the profile of bacteria inhabiting the gastrointestinal tract. The trails observed a higher proportion of *Ruminococcus* bacteria in this instance was demonstrated in the case of EEN treatment [12].

The hypothesis of the effect of a diet low in fermentable oligosaccharides, disaccharides, monosaccharides, and polyols FODMAPs was tested in the group of patients with IBD. The study revealed the potential impact of a low FODMAPs diet on the clinical response and reduction in the amount of *Bifidobacterium adolescentis*, *Bifidobacterium longum*, *Fecalibacterium prausnitz*, although a continuation is needed as the results were not statistically significant [13,14].

Fecal Microbiota Transplantation (FMT) in UC is an effective form of therapy. Statistically significant differences in response to treatment in active UC receiving FMT compared to the placebo group were demonstrated. In the first group, the positive clinical effect was associated with an increase in the microbiological diversity of the gastrointestinal tract [15].

There are no large clinical studies on the effects of microbiome modification with FMT in patients with CD so far. However, pilot studies have shown similar to that in UC, FMT therapy also led to the colonization of the recipient with the donor’s microbiota, which in turn, was conductive to maintaining remission of the disease [16].

Taking into account the abovementioned data, the diversity and specific composition of the microbiome has an impact on the development of IBD. Nevertheless, the interaction of the microbiome and intestinal epithelium/immune system is not one-sided. The host factors influence the composition of the intestinal bacterial flora and thus indirectly favor the development of IBD. The gene *C4B* is related to an early onset of IBD (pediatric inflammatory bowel disease, PIBD). Studies determining the relationship between the number of copies of the complement *C4B* gene and the activity of the complement, as well as the composition of the microbiome, have shown that the number the *C4B* gene correlated positively with an increase in the activity of the complement towards microbiome and with the escalation of dysbiosis [17].

The next mutation associated with an increased incidence of IBD early in life seems to be the *NOX1* gene mutation. NOX1 is the catalytic subunit of nicotinamide adenine dinucleotide phosphate oxidase NADPH complex 1 that generates peroxide. Localized in the membrane of intestinal epithelial cells, the enzyme is responsible for the bactericidal rupture in phagocytes. The IBD variant changes the interface between luminal microbes and epithelium. Not only these, but a significant number of other mutations have been described in people with IBD that result in the loss of intestinal integrity in genome-wide association studies (*GNA12, MUC19, XBP*) [18,19].

The tightness of the intestinal epithelium is another stage of host protection against excessive penetration of antigens from the intestinal lumen. It is conditioned by the undisturbed function of intercellular connections, including desmosomal junctions, which are related to the recruitment of desmosomal junctional protein—desmoglein 2 (DSG2), to the cell surface. The process is under control of glial cell line-derived neurotrophic factor (GDNF)—in experimental models, its reduction by the proinflammatory cytokines was associated with the loss of DSG2-mediated intercellular adhesion and impairment of the intestinal barrier function. GDF stimulation led to the restoration of epithelial function by increasing DSG2 recruitment [20]. The increase in pro-inflammatory cytokines is secondary to the loss of intestinal barrier integrity, which is influenced, by GDNF. GDNF acts through the cAMP and p38 MAPK pathways, affecting the tightness of the epithelium. The loss of GDNP activity appears to have a similar effect to that of TNF-α, which affects p38MAPK by altering epithelial properties [21]. A specific product of the interaction of the intestinal epithelium and microbiota is the mucosa that separates the external environment from the internal one. Its final composition results from the ability of humans to create its structural elements as well as from the enzymatic activity of bacteria in relation to their components.

The interaction between the host and bacteria determines mucosal integrity that supports epithelial cells. The mucosal layer consists of inner and outer parts. The outer is a natural location of the commensal flora, the inner separates them with the epithelium. The structure of the mucosal coat is formed by the large glycosylated gel-forming glycoproteins called mucins that are produced by the goblet cells. The intestinal mucosal coat is primarily composed of MUC2 and MUC5AC. The Fc Gamma Binding Protein (FCGBP), Zymogen Granule Protein 16 (ZG16), Anterior Gradient 2 (AGR2), Chloride Channel Accessory 1 (CLCA1) and Trefoil Factor 3 (TFF3) are the other ingredients of this structure. The gate-function over communication between lumen antigen is not only limited to forming a physical border by secretory parts, but also by transmembrane mucins that seem to play a role in the transduction of signal into epithelial cells (MUC 17) and further to immune cells in response to bacterial stimulation [22].

So far, a lot is known about structure and production of mucins by goblet cells, but the way the bacteria modify the mucus remains unclear. In animal models it was noted that some types of bacteria can affect mucin degradation by their impact on the enzymes crucial to glycans changeover. *Bifidobacterium dentium* has been identified as one that can colonize the mucus and secrete metabolites that upregulate goblet cell’s function and MUC2 production. It makes *B. dentium* a potential target with protective value against intestinal inflammation [23].

The role of mucins in the pathogenesis of IBD was also confirmed by another animal model. The deficit of MUC 2 mucin made the microbiota-positive animals prone to develop induced colitis while microbiota-free mice were resistant to the disease [24,25].

The data shows that the interaction of mucus parts coming from the host and secreted by bacteria may have a selective effect on the human microbial composition. This interdependence determines the maintenance of the immune balance and both a change in the mucus structure, as well as in the microbiome may increase the exposure to antigens [26,27].

Furthermore, the protective role of mucus is important not only in the prevention of IBD, but in neoplastic transformation as well. The genetic variant of specific forms of type 2 mucin has been described in colon cancer (MUC2-MS8) [28].

Once the epithelial defence line is broken, the transmission of activating signal coming from the lummen leads further through receptors on effector cells. By stimulating membrane Toll-like receptors (TLRs) and intracellular NOD receptors (nucleotide-binding oligomerization domain proteins, NOD1, NOD2) in immune cells, including antigen-presenting ones (APCs), the cytokine pathway of the cellular immune response and tissue remodelling is activated.

Genetically induced impairment of antigen identification by immune cells related to a mutation of the *NOD2* was described in CD and was the first of the discovered disease-related mutations [29,30]. This gene encodes an intracellular receptor that recognizes peptidoglycans of Gram-positive and Gram-negative bacteria—MDP (muramyl dipeptide) and is expressed in the monocyte lineage, in APCs, Paneth cells, and stem cells. The gene product, together with *NOD1*, constitutes a family of intracellular receptors-PRRs (pathogen recognition receptors), which are an important component of the antimicrobial response of the organism, acting in parallel to the activating pathway stimulated by surface pathogen recognition receptors—Toll-like receptors. The products of the *NOD2* gene are responsible for a number of defence functions. By activating the autophagy process they eliminate intracellular pathogens of the epithelial layer; by stimulating Paneth cells to produce antibacterial peptides, they reduce the invasiveness of bacteria penetrating from the intestinal lumen and by stimulating dendritic cells, they trigger a specific immune system response [31,32].

The genetic polymorphism in the *NOD2* gene determines the different degree of interaction between the gut microbiome and the host. In individuals homozygous for specific variants of the genes (missense mutations, *R702W* and *G908R*, and one frameshift mutation, *L1007fsinsC L1007fs*), a significantly higher incidence of CD associated with decreased functional integrity of the epithelium is observed. A lack of adequate activation of monocytes with the mutation of the *NOD2* gene, in response to MDP stimulation, leads to an impairment of the immune impact on the intestinal microbiome [33,34]. The lower production of human antibacterial α-defensins (HD5 and HD6) by Paneth cells with the *NOD2* mutation, in patients with CD and animals with induced enteritis, reduces the ability of the innate answer to bacterial antigens [31,35].

At the end, the disruption of the autophagy process in response to incorrect pathogen identification blocks tissue renewal and healing. The higher risk of the disease depends not only on the loss of the *NOD2* gene function but also on the presence of the pathogenic *ATG16L1-T300A* gene variant, resulting in the disturbance of autophagosome formation [36].

The primary immune response is the trigger element for the development of IBD. Its effectiveness is determined by both modifiable external factors, such as the composition of the microbiome, the consumption of antibiotics, or contamination of food and the environment with chemical substances, as well as genetically defined abnormalities in the identification of pathogens and damage to the structure of the gastrointestinal wall, including loss of tightness of intercellular junctions and disturbances in the formation of a mucin protective coat. covering the epithelium. Despite the wide knowledge of the role of primary immune response in IBD, the change of its function in prevention against IBD still remains limited.

## 3. Adaptive Immunity in Chronic Inflammation

Activation and inhibition of the adaptive immune reaction is settled by a multi-level regulated network through the expression of pro and anti-inflammatory cytokines, which determines the maintenance of homeostasis between the external and internal environment of the human body. Studies on the pathomechanism of the development of IBD, initially indicated the dominant and discriminatory role of type 1 and 2 T helper lymphocytes, in the promotion of the inflammation. Increased activity of Th1 lymphocytes in the course of CD and Th2 lymphocytes in the course of UC has been observed. Homeostatic role of Th1 lymphocytes is to control interactions with pathogens by the formation of interferon gamma (IFN-γ) and IL2, while Th2 lymphocytes by producing IL: 4, 5, 13 play a protective role against parasitic infestation. At present, the importance of Th17 lymphocytes, which constitute the main type of helper lymphocytes in the intestinal epithelium, is more emphasized. By accumulating in the lamina propria, they constitute a key element of defence against extracellular and intracellular bacteria (*Klebsiella pneumoniae*, *Listeria monocytogenes*, *Salmonella enterica*, *Mycobacterium tuberculosis* and fungi *Candidia albicans* [37]). Their activity is regulated, among others, through IL23 (activation) and IL10 (inhibition). The disturbances in response to stimulation with these cytokines, at various levels of the signal transduction pathway (interleukin deficiency, receptor defect, intracellular signal transduction defect) were found in both animal models of induced enterocolitis and in human observational studies [37,38,39].

## 4. The Group of 12/23 Cytokines

A component of the adaptive response in IBD is the stimulation of T-helper cell differentiation. Among the subpopulations of lymphocytes involved in the maintenance of the inflammation, the important role is played by Th17 lymphocytes derived from naive Th0 lymphocytes under control of IL23 and IL12 [40]. Interleukin 23 together with IL12, 27 and 35 are part of the IL12/23 group with a specific heterodimeric structure of the α and β units. They are produced by activated antigen-presenting cells [41], and by binding with the p40 subunit to the surface CD4 + receptor IL-12Rb1 that start the intracellular pathway of activation of the effector cell. The further transduction of the signal is mediated by transcriptional factors—STAT proteins (signal transducers and activators of transcription) which are regulated by enzymatic function of Janus kinases (JAK). The cytokine mediated signal transmission through JAK-STAT pathway (IL12/23-IL12Rg1-SAT-JAK) ultimately activates Th17 cells, which are the source of several pro-inflammatory cytokines: IL17A, IL17F, IL22, IL26, and the chemokine CCL20. Increased expression of these cytokines has been reported in the course of IBD [42]. Genetic mutations on each of the levels of Th17 stimulation, i.e., the IL12/23 receptor, activity of STAT proteins and Janus kinases (IL12B, JAK2, STAT3, CCR6 and TNFSF15) may promote IBD [43]. In clinical studies overexpression of the mRNA of IL17) IFN-***γ*** and IL23 receptor (IL23 R) in lamina propria CD4+ leukocytes, was found in intestinal biopsies from pathologically altered gastrointestinal sections of patients with UC and CD. Similarly, in both diseases, the production of IL17 dependent on IL23 was significantly increased. This shows the involvement of the IL23—dependent pro-inflammatory pathway in the differentiation of Th17 cells involved in the initiation and maintenance of the disease [42].

The practical dimension of the discovery of the pro-inflammatory action of IL12/23 was used in the treatment of CD and UC. Blockade with a monoclonal antibody of the p40 subunit of 12/23 interleukins, inhibited the interaction of cytokines with their surface receptor on the Th lymphocytes and proved to be an effective form of treatment leading to the remission of IBD.

## 5. Interleukin 10

Produced by the regulatory T cells (Foxp3-Tr1 cells and Fox3 + Treg cells), as well as by cells of the monocytic line, IL10 affects the activity of Th17 lymphocytes (suppressor effect) which are donors of IL17, IL23, IL6. These interleukins are activators of transcription via STAT3 and RORc. They are involved in the recruitment of neutrophils in inflamed tissue, and in the activation of mesenchymal cells that play a major role in fibrosis in CD. IL10 by binding to the IL10R1 receptor on leukocytes, uses the STAT 3 signalling pathway, regulated by the degree of phosphorylation (by the action of JAK1). The effectors for IL10—Th1 and Th2 are also its donors, which is an element of the autocrine regulation of the immune response. IL10 has the ability to inhibit the APCs by reducing the expression of major histocompatibility complex (MHC) and to control another level of adaptive immune answer. Moreover, IL10 influences the proper functioning of the intestinal epithelial barrier by modelling the renewal of intestinal stem cells and the regulation of the intestinal microflora in the fucosylation process [44].

The importance of the loss of IL10 function at various levels of its effect has been reported in human IBD. According to literature data, mutations of IL10 (loss-of-function mutation) or IL10 receptor IL10R (α and β) were confirmed in children with a severe course of IBD with onset in infancy. It has become a standard to look for the deficiency of IL10/IL10 receptors as a trigger in the case of infantile IBD. If monogenetic disease is confirmed a bone marrow/stem cell transplantation is the option for effective treatment [45,46,47,48]. However, attempts to single use of IL10 in the treatment of IBD stays limited and the results are inconclusive.

The local administration of IL10 in the treatment of severe rectal IBD have shown activation of Th2-dependent immune responses promoting healing [49]. However, in another studies the treatment with recombinant human IL-10 (rHuIL-10) administered subcutaneously, once daily for 28 days, in in vivo studies did not give the anti-inflammatory and immunosuppressive effect noted in vitro [50].

Despite presented limitations of the therapeutic use of IL10, its role in modelling the inflammatory pathway seems to be crucial. In clinical trials the role of IL10 in IBD has been tested. In the studies with IBD versus control group, overexpression of IL23 and decreased production of IL10 in intestinal tissue biopsies was confirmed. The transcription of IL10 was inhibited in CD4 (+) T cells. The observed IL23/IL10 relation was associated with a total reduction in IgA level and reduced gut barrier efficacy [51]. In Behçet’s disease that frequently comes along with intestinal manifestation, a decreased expression of IL10 mRNA related to hypermethylation of the promoter region was demonstrated [52]. These observations make more research in this field necessary.

## 6. Fibrosis

Tissue reconstruction in IBD is a result of dysregulation of the healing process and is particularly evident in CD, where a typical complication is the presence of full-wall fibrosis, leading to gastrointestinal strictures. Pro-inflammatory cytokines and growth factors secreted by activated immune and epithelial cells, stimulate mesenchymal cells including fibroblasts, myofibroblasts and smooth muscle cells, to produce extracellular matrix (ECM) at the inflamed intestines. Mesenchymal cells are the effectors of the stimulation by pro-inflammatory cytokines, they can also directly interact with intestinal lumen antigens, including PAMPs (pathogen-associated molecular patterns), dsRNA and bacterial DNA via TLRs and NOD 2 receptors. Thus, damage to the intestinal barrier that is prone to increased antigenic exposure leads to chronic stimulation of the activity of these cells. Disruption of the pathogen recognition pathway at the receptor level is another component of mesenchymal cell stimulation. Mutation of the *NOD2* gene that inhibits the elimination of the pathogen, results in constant stimulation of the immune system cells. The presence of a pathogenic variant of the *NOD2*/*CARD15* gene in a patient CD increases the risk of a severe course of the disease 10-fold. Additionally, other mutations affecting the ability to identify and model the gut microbiome, including ATG16LI related to the autophagy process, increase the risk of pathological tissue remodeling [19,53]. All of these mechanisms are related to the microbiota-host interaction.

Among the growth factors that model fibrosis, the main role is played by transforming growth factor beta (TGF-β, TGF-β1, TGF-β3). TGF-β activates all types of mesenchymal cells, stimulating them to produce extracellular matrix components, including type I collagen, which constitutes about 70% of intestinal collagen. Interleukin 17, produced by Th17 lymphocytes under IL23-IL23R stimulation, is another cytokine that activates mesenchymal cells for ECM remodeling. Similarly, IL1, IL6 and IL13 have stimulating properties for mesenchymal cells. The mesenchymal cells, apart from paracrine stimulation from immune cells, are subject to autocrine regulation, and by producing zinc metalloproteinases—MMPs (matrix metalloproteinases) and inhibitors of these metalloproteinases—TIMPs (tissue inhibitors of matrix metalloproteinases), they balance the effects of self-activity. Disturbances in this balance, i.e., an increase in the level of MMPs (MMPs 1, 2, 3 and 9) in relation to TIMPs, have been described in IBD both in tissue biopsies and in serum [54,55,56,57].

Matrix metalloproteinases are released in the form of proenzymes by mesenchymal cells as well as by immune cells, i.e., T lymphocytes, cells of the granulocytic lineage: neutrophils, macrophages, eosinophils. Proenzymes are activated by the tissue plasminogen activator, urokinase. Their expression is controlled at the level of transcription by cytokines such as tumor necrosis factor alpha (TNF-α) and IL1 [58,59]. Under biological equilibrium conditions, these cytokines indirectly control tissue healing by regulating the activity of metalloproteinases. Immune system hyperactivation manifested by an increase in the activity of cytokines with a pro-inflammatory profile, such as TNF-α or IL1 produced by the mesenchymal cells as a result of stimulation of the Il23-IL23R-IL17 pathway, promote the degradation of the extracellular matrix, increasing pathological tissue remodeling leading to acute and chronic damage to the tissues of the gastrointestinal tract in the course of IBD [60].

The activation of mesenchymal cells is another example of multilevel stimulation by both direct interaction with the gut microbiome and stimulation of adaptive cytokine-mediated immune cells response. The complexity of the system maintains a delicate balance in response to environmental stimuli, and enables the healing of damaged tissues, which is a condition for maintaining the proper barrier capacity of the intestinal epithelium. A failure of this mechanism may affect various components of the cascade, resulting in phenotypically different forms of IBD. The progress of research on the pathways of activation and inhibition of the immune reaction, as well as tissue remodeling controlled by the immune response, creates the possibility of influencing the evolution of autoimmune diseases and limiting the serious complications of a long-term process.

## 7. Conclusions

Is the etiology of IBD still a mystery? The answer to this question remains problematic. The multilevel control of the immune response makes it difficult to identify the triggering factor for the persistence of inflammation in the intestinal wall. The interaction between lumen antigens and the epithelial border defense line appears to be the main factors in induction, but the further transmission of primary signals modulated by genetic predisposition is no less important. The interesting issue, what determines the development of a particular type of IBD, is still unanswered. The pattern of the microbiome, by activating distinct signaling pathways, seems to be significant. It has been postulated that e. g. viral infections may regulate genes expression promoting CD, however the involvement of specific pathogens has not been established [61]. The role of microRNA (miRNA) in conditioning CD or UC development has also been indicated. In the Schaefer et al. [62] study, in the colon biopsy miR-146a was elevated in UC while miR-19a, miR-31, miR-142-3p, miR-375, and miR-494 were CD specific. Nevertheless, the main mechanisms of the immune response seem to be common to different types of IBD, and treatment of both diseases by blocking the pro-inflammatory factors has been shown to be effective in both CD and UC.

In clinical practice, the knowledge coming from the understanding of the mechanisms of mutual dependence of structural elements and cytokine mediators regulating their activity, has been used in the creation of new drugs, effective in IBD. Inhibition of TNF-α by the therapeutic monoclonal antibodies (infliximab, adalimumab) proved to be a breakthrough in the treatment of IBD for both, CD and UC, changed patients’ prognosis. Further antibodies, including those targeting the inhibition of the inflammatory pathway dependent on the activation of IL23-Il23R (ustekinumab), made it possible to obtain clinical effects, especially in patients with moderate and severe course of the inflammatory process. Currently, clinical trials are underway to combine monoclonal antibodies with different mechanisms of action, e.g., anti TNF-α and anti IL12/23, to obtain a response in severe patients who have not achieved remission on monotherapy with a biological agent. 

Other drugs effectively used in the treatment of IBD, which have the ability to target the immune system response modulation, include Janus Kinases Inhibitors and antibodies that inhibit the migration of leukocytes to the gastrointestinal tract, such as vedolizumab (anti-α4 β7 integrins), abrilumab (anti-α4β7 IgG2), etrolizumab (anti-β7) or sphingosine-1 Phosphate Receptor Modulators. The development of new forms of therapy was possible thanks to the recognition of the interdependencies regulating the final immune response, and further research seems to be promising in the context of individualizing treatment and reducing the general effect of previously used drugs [63,64,65,66]. The scheme of IBD pathogenesis and the mechanisms of action of biological drugs in IBD patients are presented in Figure 1 and Figure 2.

## Figures and Tables

**Figure 1 ijms-23-12445-f001:**
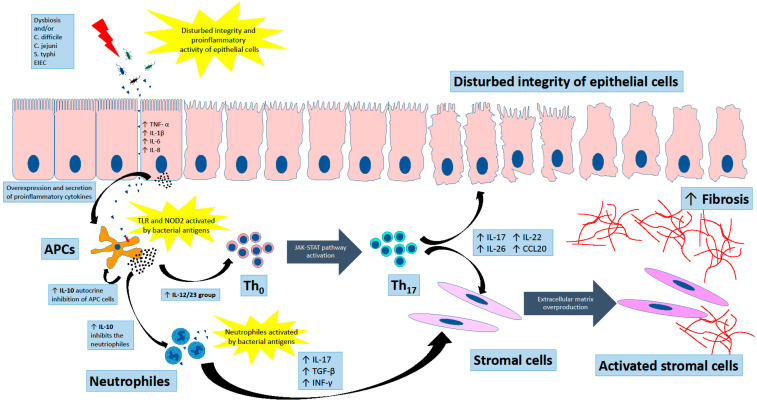
Pathogenesis of inflammatory bowel diseases (IBDs). The dysbiosis of gut microbiota is a cause of exacerbation of the disease and determines the severity of the mucous inflammation. It causes the imbalance in the integrity of epithelium barrier and promotes infection of, i.e., *C. jejuni*, *S. typhi*, enteroadherent or serotypes enteroinvasive of *E. coli* (EIEC). By stimulating membrane Toll-like receptors (TLR) and intracellular NOD receptors (nucleotide-binding oligomerization domain proteins, NOD1, NOD2) in immune cells, including antigen-presenting ones (APCs), the cytokine pathway of the cellular immune response and tissue secondary remodeling is activated. The important role is played by Th17 lymphocytes derived from naive Th0 lymphocytes under control of IL23 and IL12. The cytokine mediated signal transmission through JAKSTAT pathway ultimately activates Th17 cells, which are the source of several pro-inflammatory cytokines: IL17A, IL17F, IL22, IL26, and the chemokine CCL20. In addition, the IL-10 affects the activity of Th17 lymphocytes inhibition which are donors of IL17, IL23, IL6. As well as the IL10 has the ability to inhibit the APCs by reducing the expression of major histocompatibility complex (MHC) and to control another level of adaptive immune answer. The IL17, IL23, IL6 and TGF-β, INF-γ are involved in the recruitment of neutrophils in inflamed tissue and in the activation of mesenchymal cells that play a major role in fibrosis in IBD.

**Figure 2 ijms-23-12445-f002:**
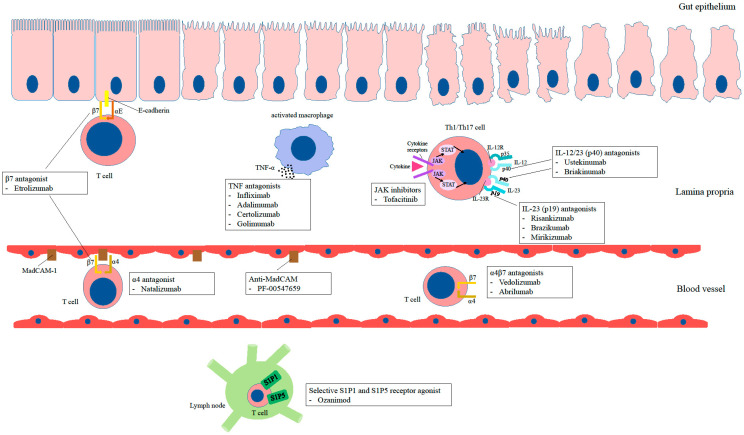
Effect of biological drugs on the modulation of the immune response in patients with inflammatory bowel diseases. TNF—tumor necrosis factor, JAK—Janus kinase, MadCam-1—Mucosal vascular addressin cell adhesion molecule 1, S1P1—Sphingosine-1-phosphate receptor 1, S1P5—Sphingosine-1-phosphate receptor 5.

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
