# Peer review of "Etiology of IBD—Is It Still a Mystery?"

_ijms, 2022, doi:10.3390/ijms232012445_

Round 1

Reviewer 1 Report (New Reviewer)

Your work improves researchers' understanding of inflammatory bowel disease,which aids in clinical disease prevention and drug development.Dysregulation of gut microbiota plays an important role in the progression of inflammatory bowel disease.Your research work should increase some research on the pathogenic mechanism of gut microbiota.Although both ulcerative colitis and Crohn's disease are inflammatory bowel diseases, there are differences in the pathogenesis and choice of treatment strategies.Should separate etiology studies be conducted for the two diseases? There have been many studies on the etiology of Crohn's disease. What is the innovation of your research?

Author Response

Dear Reviewer,

thank you for your comments and valuable suggestions improving the quality of the manuscript.

During the revision of the manuscript, we took into account most of the suggested changes.

In response to the reviewers

- the manuscript has been supplemented with a description of the mechanism of the influence of intestinal pathogens on the development of IBD (suggestion from reviews 1 and 2). 

- a paragraph on the differences in the etiopathogenesis of IBD subtypes (CD and UC) was added, however, due to the lack of breakthrough reports indicating the existence of significant differences in the immune response and its modeling factors, it was not discussed extensively. (suggestion from review 1)

- figures showing the mechanisms of action of drugs used in the treatment of IBD at various levels of activation of the inflammatory process have been added, which, as suggested by the reviewer, improves the readability and value of the manuscript. The team of Kofla-Dłubacz et al. published a complete study of current therapies in a separate manuscript - a reference in the manuscript (suggestion  from review 2)

- we included additional information about the mechanisms by which bacteria such as Campylobacter jejuni, Salmonella typhi, enteroinvasive and enteroadherent Escherichia coli serotypes, the Mycobacterium group trigger IBD pathogenesis, and the relationship between GDNF and proinflammatory cytokines (suggestion  from review 2)

- grammatical and formatting errors have been corrected

Most of the changes made in the text have been marked in yellow or pink.

We hope that You will be satisfied with the changes we have made.

Reviewer 2 Report (New Reviewer)

Manuscript Summary:

Inflammatory bowel disease (IBD) is a chronic inflammatory disease of the gastrointestinal tract, which clinically contains Crohn's disease, ulcerative colitis (UC), and other conditions. The inflammation of the intestinal mucosa in IBD is characterized by abdominal pain, diarrhea, bloody stools, weight loss, and the influx of neutrophils and macrophages that produce cytokines, proteolytic enzymes, and free radicals, resulting in inflammation and ulceration. Considerable progress over decades has demonstrated that the etiology of IBDs is multifactorial and is associated with genetic susceptibility of the host, intestinal microbiota, other environmental factors, and immunological abnormalities. In this review, Kofla-Dłubacz et al. describe the primary and adaptive immune system responses in the gut and the known immune pathogenetic pathways leading to the development of IBD.

Overall, this manuscript is perfectly suited for the scope of the International Journal of Molecular Sciences. Furthermore, this manuscript is written in a manner that is appealing to a wide range of audiences and crisply covers a  breadth of information, especially the consistent linkage to genotype-phenotype correlations with the pathogenesis of IBD.

The following are my major and minor comments.

Major Comments:

While the authors briefly discuss the currently approved therapies for the treatment of IBD, if the authors were to compile a table detailing different therapeutic strategies ( based on mechanisms targeted, such as the modulation of the microbiome, modulation of the immune effectors, modulation of tolerance pathways, and modulation of fibrosis) that are either already approved by the FDA/EUA or are being evaluated in various clinical trials (including the phases of the clinical trials as well), if there are any specific patient subpopulations being targeted for specific therapies (personalized medicine approaches), and their respective outcomes, it would go a long way in demonstrating which therapeutic strategies are proving to be beneficial in the clinic (also showing translation from murine models to patients), thereby answering the question whether the etiology of IBD is as much a mystery even today.

Minor Comments:

1. Line 30: What is the global prevalence of IBD?

2. Lines 78 - 81: It would be worth briefly elaborating on the mechanisms by which bacteria such as Campylobacter jejuni, Salmonella typhi, enteroinvasive and enteroadherent serotypes of Escherichia coli, Mycobacterium group: (M. avium and M. paratuberculosis) trigger the pathogenesis of IBD.

3. Line 115: Awkward sentence structure. The sentence structure beginning with "However, pilot studies..." seems a bit off. Perhaps, should it say "However, pilot studies have shown similar to that in UC, FMT therapy also led to the colonization of the recipient with the donor's microbiota, which in turn, was conducive to maintaining remission of the disease"?

4. Numerous typos, grammatical and formatting errors:

- Clinical 'trials' (line 101), GDNF (line 142)

- Hyphen before the beginning of a new sentence instead of a space (line 143).

- The mentions of gene names should be italicized (e.g., line 122 - gene C4B; line 128 - NOX1)

- Do the authors mean 'XBP' (line 134)?

- Line 147: The interaction between the host and bacteria (missing 'between the')

- Line 150: Perhaps a better way to frame this sentence would be "The structure of the mucosal coat is formed by the large glycosylated gel-forming glycoproteins called mucins that are produced by the goblet cells. The intestinal mucosal coat is primarily composed of MUC2 and MUC5AC."

- Line 159: 'In animal models'

- Line 162: 'goblet cell's (missing apostrophe) function'

- Line 173: 'Furthermore,' (comma after furthermore)

- Line 182: "Function" (stricken through) should be deleted

- Line 252: STAT is misspelled

- Line 256: inconsistent spacing

- Line 271: Foxp3 misspelled

- Line 273: Awkward sentence structure. It would be better to state 'IL10 suppresses the activity of Th17 lymphocytes, which contributes to the pool of IL17, IL23, and IL6 cytokines'

- Lines 276-277: Confusing sentence structure. An alternative version would be 'IL10 by binding to the IL10R1 receptor on leukocytes signals via the JAK1-STAT3 pathway, which is regulated by the degree of JAK1-mediated STAT3 phosphorylation.'

- Multiple instances, e.g., line 278: "Donor" is quite an awkward usage for a cytokine-producing cell type.

4. Line 140: It would be worth specifying which proinflammatory cytokines are associated with reduced GDNF levels.

5. Line 272: It is worth calling out the myeloid populations that produce IL-10

Author Response

Dear Reviewer,

thank you for your comments and valuable suggestions improving the quality of the manuscript.

During the revision of the manuscript, we took into account most of the suggested changes.

In response to the reviewers

- the manuscript has been supplemented with a description of the mechanism of the influence of intestinal pathogens on the development of IBD (suggestion from reviews 1 and 2). 

- a paragraph on the differences in the etiopathogenesis of IBD subtypes (CD and UC) was added, however, due to the lack of breakthrough reports indicating the existence of significant differences in the immune response and its modeling factors, it was not discussed extensively. (suggestion from review 1)

- figures showing the mechanisms of action of drugs used in the treatment of IBD at various levels of activation of the inflammatory process have been added, which, as suggested by the reviewer, improves the readability and value of the manuscript. The team of Kofla-Dłubacz et al. published a complete study of current therapies in a separate manuscript - a reference in the manuscript (suggestion  from review 2)

- we included additional information about the mechanisms by which bacteria such as Campylobacter jejuni, Salmonella typhi, enteroinvasive and enteroadherent Escherichia coli serotypes, the Mycobacterium group trigger IBD pathogenesis, and the relationship between GDNF and proinflammatory cytokines (suggestion  from review 2)

- grammatical and formatting errors have been corrected

Most of the changes made in the text have been marked in yellow or pink.

We hope that You will be satisfied with the changes we have made.

This manuscript is a resubmission of an earlier submission. The following is a list of the peer review reports and author responses from that submission.

Round 1

Reviewer 1 Report

The authors of the review: Etiology of IBD- is it still a mystery? describe what is known in brief about the role of microbiome, immune system and interleukin profiling in the initiation and progression of IBD. 

The review provides valuable however redundant information with no major addition to existing reviews. Below are some recommendations and comments: 

1- Please confirm stats Line 25-27. Appears to be inconsistency in the numbers.

2- The abstract is barely indicative about what the article is addressing

3- Extensive English Editing: Majority of the article is hard to follow up due to broken structure and incomplete sentences. 

4- Line 123: MDP is Muramyl Dipeptide / Have not came across Muracyl nomenclature

5-Line 147: Autophagy process

6- Paragrahps 3 and 4 have redundant information

7- Much more is known about new clinical trials and IBD. Limited information is provided in this review

Reviewer 2 Report

In the entitled manuscript, “Etiology of IBD- is it still a mystery?”.  The authors seem to have attempted to write a review to summarize the immune pathogenesis of IBD and clinical therapies based upon current understandings of immune pathogenesis.  However, the writing itself doesn’t reach the level that I believe would allow the average reader to follow along easily.  Grammar errors abound in the article, especially in the first two pages (which I find a bit odd, to be honest).  Second, it is unclear where the mystery of IBD is revealed.  For example, subsection 2 is titled “Promotion of the inflammatory response in [IBD]”, and yet does not appear (if, say, one strips out all but the topic sentences of each paragraph) to discuss anything of the sort.  Instead, the section discusses the microbiome, mucins and receptors etc..  Subsections 3-5 seem to represent a discussion of adaptive immune responses in IBD, but I don’t see the logic underlying the choice of the specific three sections (titled, respectively, “Adaptive immune response in [IBD]”, “The group of 12/23/ cytokines”, and “Interleukin-10”).  And immunotherapeutic effects of anti-TNF-γ are discussed in the section on Fibrosis.  Third (and minor), no microbe names are italicized in the paper and many statements in the text are not referenced.

IBD, immunopathogenesis and immune therapies are indeed important, but have been well addressed in many review articles (see for example, PMCID: PMC8386758, 2021). When such a subject is well established and not in need of an update (which a review article might then summarize), the scientific value of a simple repetition is missing. One possible way to improve this submission might be to narrow its focus greatly, say to mesenchymal cells alone, or to the epithelial barrier alone, or even to the anti-inflammatory effects of interleukin-10.  In any event, I cannot express support for publication of this submission in its current form.